# Effectiveness of a Self-Decontaminating Coating Containing Usnic Acid in Reducing Environmental Microbial Load in Tertiary-Care Hospitals

**DOI:** 10.3390/ijerph20085434

**Published:** 2023-04-07

**Authors:** Helena C. Maltezou, Nikolaos Papamichalopoulos, Elina Horefti, Maria Tseroni, Amalia Karapanou, Maria N. Gamaletsou, Lamprini Veneti, Anastasios Ioannidis, Marina Panagiotou, Evangelia Dimitroulia, Antonios Vasilogiannakopoulos, Emmanouil Angelakis, Stylianos Chatzipanagiotou, Nikolaos V. Sipsas

**Affiliations:** 1Directorate of Research, Studies, and Documentation, National Public Health Organization, 15123 Athens, Greece; 2Department of Medical Biopathology, Aeginition Hospital, Medical School, National and Kapodistrian University of Athens, 72–74 Vas. Sophias Ave, 11528 Athens, Greece; 3Diagnostic Department and Public Health Laboratories, Hellenic Pasteur Institute, 127 Vas. Sophias Ave, 11521 Athens, Greece; 4Directorate of Epidemiological Surveillance for Infectious Diseases, National Public Health Organization, 15123 Athens, Greece; 5Infection Control Committee, Laiko General Hospital, 11527 Athens, Greece; 6Department of Pathophysiology, Laiko General Hospital, Medical School, National and Kapodistrian University of Athens, 11527 Athens, Greece; 7Independent Researcher, 11744 Athens, Greece; 8Department of Nursing, Faculty of Health Sciences, University of Peloponnese, 22100 Tripoli, Greece; 9Infection Control Committee, Henry Dunant Hospital Center, 11526 Athens, Greece

**Keywords:** usnicacid, environmental contamination, hospital, surfaces, SARS-CoV-2, gram-negative bacteria

## Abstract

Surfaces have been implicated in the transmission of pathogens in hospitals. This study aimed to assess the effectiveness of an usnic-acid-containing self-decontaminating coating in reducing microbial surface contamination in tertiary-care hospitals. Samples were collected from surfaces 9 days before coating application, and 3, 10, and 21 days after its application (phases 1, 2, 3, and 4, respectively). Samples were tested for bacteria, fungi, and SARS-CoV2. In phase 1, 53/69 (76.8%) samples tested positive for bacteria, 9/69 (13.0%) for fungi, and 10/139 (7.2%) for SARS-CoV-2. In phase 2, 4/69 (5.8%) samples tested positive for bacteria, while 69 and 139 samples were negative for fungi and SARS-CoV-2, respectively. In phase 3, 3/69 (4.3%) samples were positive for bacteria, 1/139 (0.7%) samples tested positive for SARS-CoV-2, while 69 samples were negative for fungi. In phase 4, 1/69 (1.4%) tested positive for bacteria, while no fungus or SARS-CoV-2 were detected. After the coating was applied, the bacterial load was reduced by 87% in phase 2 (RR = 0.132; 95% CI: 0.108–0.162); 99% in phase 3 (RR = 0.006; 95% CI: 0.003–0.015); and 100% in phase 4 (RR = 0.001; 95% CI: 0.000–0.009). These data indicate that the usnic-acid-containing coating was effective in eliminating bacterial, fungal, and SARS-CoV-2 contamination on surfaces in hospitals.Our findings support the benefit ofan usnic-acid-containing coating in reducing the microbial load on healthcare surfaces.

## 1. Introduction

According to the World Health Organization (WHO), 700,000 deaths globally are associated with multidrug-resistant (MDR) infections, and this number will increase to 10million by 2050 if appropriate interventions are not implemented [1]. Greece is among the European countries with the highest rates of healthcare-associated infections and antimicrobial resistance in hospitals [2,3,4].

Contaminated surfaces have been often traced as sources of transmission of infections in healthcare facilities [5,6]. Several bacteria, including MDR bacteria (e.g., *Klebsiella pneumoniae,* methicillin-resistant *Staphylococcus aureus* (MRSA), vancomycin-resistant enterococci (VRE)), may survive on surfaces in healthcare facilities, colonize the hands of healthcare personnel and the flora of patients, especially those undergoing interventions, and contribute to the spread of infection [5,7,8,9,10,11]. Viruses, including influenza viruses, severe acute respiratory syndrome coronavirus 2 (SARS-CoV-2), and norovirus, are also detected on environmental surfaces and can be transmitted through contaminated surfaces [5,12,13,14,15]. Under specific temperature and humidity conditions, SARS-CoV-2 can survive on surfaces and remain infectious for several days [16]. Frequent hand hygiene, routine cleaning, and disinfection of nosocomial environments are key measures to protect patients and healthcare personnel, particularly in healthcare settings with high-risk patients such as intensive care units (ICUs) [6,7,14,15,17,18]. Nevertheless, long-term strategies to eliminate environmental contamination in healthcare facilities are needed, with an emphasis on settings where high-risk patients receive healthcare [1].

Usnic acid, a dibenzofuran derived from lichens, and its derivatives possess antibacterial activity against several Gram-positive and Gram-negative healthcare-associated bacteria, such as *Staphylococcus aureus*, *S. epidermidis*, MRSA, *Pseudomonas aeruginosa*, VRE, and *Escherichia coli;* usnic acid and its derivatives also demonstrate good antifungal activity against *Candida* spp. and *Aspergillus* spp. [19,20,21,22,23,24,25,26]. These activities are attributed to the disruption of cell membranes, but also to the inhibition of biofilm and the prevention of adhesion, while on many occasions, these activities are comparable to or even higher than those of several antimicrobial agents [20,21,22,23]. In addition, usnic acid and its derivatives demonstrate activity against viruses of importance for healthcare settings such as influenza virus, norovirus, and SARS-CoV-2, which is conferred through the suppression of virus replication [27,28].

The natural compound of usnic acid has been incorporated into a polymer, resulting in the formation of a semi-liquid, self-decontaminating material named Natural Protective Shield 360° (NPS 360°; hereafter referred to as coating). The coating serves as an antimicrobial barrier [29]. Because its antimicrobial activity is attributed mostly to mechanical factors, there is no risk for the emergence of antimicrobial resistance. This characteristic, along with its prolonged duration of activity, renders the coating an excellent strategy for long-term infection control for healthcare facilities, particularly in settings with high disinfection needs, such as ICUs [29]. The coating is not hazardous, according to Regulation 1272/2008 of the European Parliament [29].

To our knowledge, there are no published data on the effectiveness of the application of the usnic-acid-containing self-decontaminating coating NPS 360° in healthcare facilities. This study aimed to estimate the effectiveness of the coating in reducing microbial surface contamination in tertiary-care hospitals in Greece.

## 2. Material and Methods

### 2.1. Setting

The study was conducted in a 539-bed tertiary-care hospital (Hospital A) and a 462-bed tertiary-care hospital (Hospital B) in Athens. Cleaning and disinfection are performed twice daily in Emergency Departments, public areas, and rooms, and 3–4 times daily in ICUs and other high-risk areas using chlorine- or alcohol-containing products (tablets, liquids, sprays). Public areas and frequently touched surfaces in Emergency Departments and ICUs are decontaminated with bleach- or H_2_O_2_-soaked cloths every two hours.

### 2.2. Application of the Coating 

On 13 May 2022, the coating was applied to a variety of pre-selected surfaces of the Emergency Departments, ICUs, COVID-19 patients’ rooms, Radiology and CT Departments, stairs, and elevators of both hospitals, in accordance with the manufacturer’s instructions. All sampled sites operated a few hours later, with no disruption of healthcare services.

### 2.3. Environmental Investigation 

Two trained healthcare professionals conducted the environmental sampling. In particular, there were four sampling phases: sampling phase 1 (9 days before the coating was applied; 4 May 2022), sampling phase 2 (3 days after the coating was applied; 16 May 2022), sampling phase 3 (10 days after the coating was applied; 23 May 2022), and sampling phase 4 (21 days after the coating was applied; 3 June 2022) (Figure 1). 

A total of 2samples were collected from 69 pre-selected surfaces per sampling phase in Hospital A (for detection of bacteria/fungi and SARS-CoV-2, each), while 1sample was collected from 70 pre-selected surfaces per sampling phase in Hospital B (for detection of SARS-CoV-2), adding up to 208 surface samples per phase. Samples were collected from 9.00 a.m. to 1.00 p.m. while hospitals were on duty but before routine cleaning and disinfection. Sampling surfaces included a wide range of materials, such as metallic objects (e.g., trolleys, stairs’ handrails), non-porous plastic/resin objects (e.g., elevators’ buttons, counters), and non-porous leather objects (e.g., stretchers). Sampling was conducted using two sterile swabs in 2 mL of viral nucleic acid sample preservation. Each sampled surface area was approximately 100 cm^2^ (10 cm × 10 cm). The EN ISO 18593:2018 was followed for sampling procedures. Sampling was repeated on 16 May 2022 (sampling phase 2), on 23 May 2022 (sampling phase 3), and on 3 June 2022 (sampling phase 4), as on 4 May 2022 (sampling 1), adding up to 832 surface samples.

### 2.4. Laboratory Testing 

Samples from Hospital A were transferred to the Aeginition Hospital Department of Biopathology (University of Athens Medical School) for testing, according to reference [30]. In particular, bacteria and fungi were identified with aMicroScan autoSCAN-4 System, Beckman Coulter. The swabs were broken and submerged in a microtube containing 2.5 mL of sterilized buffered peptone water as elution medium. The microtubeswere vortexed for 20 s. An amount of 1ml of supernatant was tested for further quantitative analyses. Elusion medium swabs were cultured on conventional solid media for a wide range of Gram-positive and Gram-negative bacteria and fungi: blood agar, chocolate agar, MacConkey agar, Sabourauddextrose agar, mannitolsalt agar (Chapman), and *Clostridioides difficile* agar. For each swab, 200 μL of elution medium was streaked on a whole agar plate. Colonies were counted after incubation for 24–48 h. Incubation was performed at 37 °C under aerobic conditions for all culture media, except for *C. difficile*, which was incubated at 37 °C under anaerobic conditions. Culture results were presented in colony-forming units (CFUs)/100 cm^2^. The total bacterial presence was considered since we evaluated bacterial growth as an indicator of general cleanliness. Culture media were manufactured according to ISO 9001:2015–ISO 13485:2016 with CE Mark and were supplied by Bioprepare Microbiology, Keratea—Attica, Greece.

Samples for the detection of SARS-CoV-2 were collected in Viral Transport Medium and were transported in cooler bags in order to maintain integrity. Samples from Hospital A were tested for SARS-CoV-2 at the Hellenic Pasteur Institute (Athens). Total RNA was extracted using aMagCore^®^ Viral Nucleic Acid Extraction kit 203. Viral RNA was detected by the qualitative real-time reverse transcription (RT)-PCR, for the viral envelope protein (E)-encoding gene and the RNA-dependent RNA polymerase (RdRp) gene [31]. The Virus Transport Medium (GLYE), CE-IVD, was provided by BioprepareMicrobiology. Samples from Hospital B were tested for SARS-CoV-2 at the Diagnostic Laboratories of the hospital, as described elsewhere [32].

### 2.5. Statistical Analysis

Categorical variables were depicted as percentages. For continuous variables, we estimated the median and range. Poisson regression was applied for the comparison of bacterial growth after the intervention (comparison of the number of bacteria in sampling phase 1 separately with sampling phases 2, 3, and 4). Additionally, *p*-values of ≤0.05 were considered statistically significant. STATA software (Stata Corporation, College Station, TX, USA, version 17) was used for the statistical analysis.

## 3. Results 

The results of the environmental investigation for bacteria, fungi, and SARS-CoV-2 are shown per sampling phase in Table 1, Table 2 and Table 3, respectively. 

Regarding positivity rates per hospital sector, samples collected from COVID-19 rooms, stairs, elevators, and Radiology and CT Departments had the highest detection rates (range: 10.0–10.2%), while samples collected from ICUs and Emergency Departments had lower detection rates (7.1% and 7.6%, respectively). 

### 3.1. Sampling Phase 1

Before the coating was applied, 53 (76.8%) of 69 samples collected from Hospital A tested positive for bacteria and/or fungi (Table 1 and Table 2). The bacteria and fungi detected in Hospital A are listed in Appendix A by sampling surface and phase.

Coagulase-negative Staphylococcus (CNS) was the most frequent bacterium detected (39 samples), followed by *Corynobacterium* spp. (18 samples), *S. aureus* (15 samples), and *Micrococcus* spp. (9 samples). Bacteria were detected in the following hospital sectors: stairs and elevators (13 positive samples), ICU (12 positive samples), Emergency Department (12 positive samples), Radiology and CT Departments (10 positive samples), and COVID-19 patients’ rooms (6 positive samples). *Aspergillus* spp. (mainly *A. niger*) was the prevalent fungus detected (five samples). Overall, fungi were detected in the Emergency Department, the ICU, the Radiology and CT Department, and the stairs and elevators in Hospital A (three, three, one, and two positive samples, respectively). In total, staphylococci were detected in 49 (71%) of 69 samples. The median CFU/100 cm^2^ for bacteria was 4 (range: 0–74, IQR:12).

Moreover, before the application of the coating, 2 (2.9%) of 69 samples collected from Hospital A tested positive for SARS-CoV-2, and 8 (11.4%) of 70 samples collected from Hospital B tested positive for SARS-CoV-2 (Table 3). In particular, SARS-CoV-2-positive samples were collected from COVID-19 patients’ rooms and the Radiology and CT Department of Hospital A (one positive sample each) and from the Emergency Department, COVID-19 patients’ rooms, Radiology and CT Department, and stairs and elevators of Hospital B (three, two, one, and two positive samples, respectively).

### 3.2. Sampling Phase 2

Three days after the coating was applied in Hospital A (sampling phase 2), 4 (5.8%) of 69 samples tested positive for bacteria (2 from the ICU and 2 from the Radiology and CT Department); CNS grew in all 4 samples and *Micrococcus* spp. in 2 (Appendix A). All samples tested negative for fungi and SARS-CoV-2. The median CFU/100 cm^2^ for bacteria was 0 (range: 0–74, IQR:0). All 70 samples collected from Hospital B in sampling phase 2 tested negative for SARS-CoV-2. 

### 3.3. Sampling Phase 3

Ten days after the application of the coating (sampling phase 3) in Hospital A, 3 (4.3%) of 69 samples were positive for bacteria (CNS in 2 and *Micrococcus* spp. in 1). The positive samples were collected from the ICU and stairs and elevators (two and one positive samples, respectively). SARS-CoV-2 was detected in 1 (1.4%) of 69 samples collected from Hospital A. The positive sample was collected from the stairs and elevators of Hospital A. No fungus was detected at that time. The median CFU/100 cm^2^ for bacteria was 0 (range: 0–3, IQR:0). In addition, all (70) samples collected from Hospital B in sampling phase 3 tested negative for SARS-CoV-2. 

### 3.4. Sampling Phase 4

Sampling phase 4 yielded *Micrococcus* spp. in 1 (1.4%) of 69 samples collected in Hospital A. The positive sample was collected from the ICU. No fungus or SARS-CoV-2 was detected at that time. The median CFU/100 cm^2^ for bacteria was 0 (range: 0–1, IQR:0). Lastly, all samples collected from Hospital B in this phase were negative for SARS-CoV-2. 

### 3.5. Antimicrobial Effectiveness of the Coating 

After the application of the coating, the bacterial load was reduced by 87% in phase 2 (RR = 0.132; 95% CI: 0.108–0.162; *p*-value < 0.001); 99% in phase 3 (RR = 0.006; 95% CI: 0.003–0.015; *p*-value < 0.001); and 100% in phase 4 (RR = 0.001; 95% CI: 0.000–0.009; *p*-value < 0.001). After the application of the coating (phases 2,3, and 4) all samples tested negative for fungi, and only one sample was positive for SARS-CoV-2 RNA. These findings indicate the antimicrobial effectivity of the coating. 

## 4. Discussion

Self-decontaminating coatings have been reportedly applied in healthcare facilities with clear advantages over regular surfaces with routine cleaning and disinfection [6,33]. To the best of our knowledge, this is the first study to generate real-life data on the effectiveness of an usnic-acid-containing self-decontaminating coating in reducing surface contamination in healthcare facilities. A large number of frequently touched surfaces in various areas in two hospitals (patient, clinical, and public spaces) were investigated. In line with in vitro studies [34,35,36], the coating was very effective in reducing the contamination of surfaces with bacteria, fungi, and SARS-CoV-2. High effectiveness against bacteria, fungi, and SARS-CoV-2 surface contamination of the same usnic-acid-containing coating was also shown in a similarly designed study that our group conducted in the Athens underground metro [37]. These findings indicate that the application of the coating may constitute an efficient strategy to eliminate surface contamination in healthcare facilities and other settings. 

Before the application of the coating, 76.8% of surface samples collected from Hospital A were contaminated with bacteria and/or fungi, indicating gaps in routine cleaning and disinfection practices and/or compliance of healthcare personnel with infection control measures. For healthcare facilities, there are no established thresholds and, therefore, no consensus regarding the acceptable number of microorganisms on surfaces. A cut-off of less than 5 CFU/cm^2^ has been proposed for clean hospital surfaces or even less than 2.5 CFU/cm^2^ by others [6]. However, for microorganisms such as *S. aureus*, VRE, *Enterobacteriae*, or *C. difficile*, a limit of less than 1 CFU/ cm^2^ indicates the need for cleaning and disinfection because they represent a real risk for infection [6]. In our study, there was a median bacterial growth of 4 CFU/100 cm^2^, with values up to 74 CFU/10^2^ cm^2^. Staphylococci prevailed on hospital surfaces, followed by *Corynobacterium* spp. and *Micrococcus* spp., while on several occasions, *Aspergillus* spp. was detected. Limits for microorganisms on hospital surfaces should be established, and a consensus is needed [6]. 

In our study, higher positivity rates were found in samples collected from Emergency Departments, which can be explained by the overcrowded waiting rooms [38]. Our findings indicate that routine cleaning and disinfection procedures did not suffice, and audits should be performed. Bacteria and fungi were also detected in many samples collected from stairs and elevators, which can be attributed to their continuous use by patients, healthcare personnel, and visitors.

In our study, 11.4% of 70 samples collected from Hospital B before the application of the coating tested positive for SARS-CoV-2. The highest positivity rates regarding SARS-CoV-2 were found in samples collected from COVID-19 patients’ rooms. SARS-CoV-2 RNA has been frequently detected in surfaces surrounding COVID-19 patients [39]; however, detection does not imply infectiousness [40]. Previous studies have shown that the virus is able to survive on various porous and non-porous surfaces, such asglass, steel, or certain types of plastic [41], and for up to 28 days at 20 °C [42].

The strength of the current study is the use of real-life data in order to assess the effectiveness of the usnic-acid-containing coating in reducing contamination in healthcare settings. For this reason, a large number of surface samples from two hospitals were tested. A limitation is that samples for the detection of bacteria and fungi were collected from Hospital A only. Another potential limitation is that the subsequent collection of two samples from each surface could limit the microbial load on the tested surface. Lastly, sampling was not performed beyond 21 days after the coating was applied. A follow-up study will be necessary to demonstrate the antimicrobial life-time of the coating. 

## 5. Conclusions 

This is a study using real-life data collected from two tertiary-care hospitals to estimate the effectiveness of a self-decontaminating coating containing usnic acid in healthcare settings. The application of the coating in the two tertiary-care hospitals proved effective in reducing bacterial, fungal, and SARS-CoV-2 contamination ona wide range of surfaces. These data support the benefit of the usnic-acid-containing coating in reducing the microbial load on healthcare surfaces.

## Figures and Tables

**Figure 1 ijerph-20-05434-f001:**
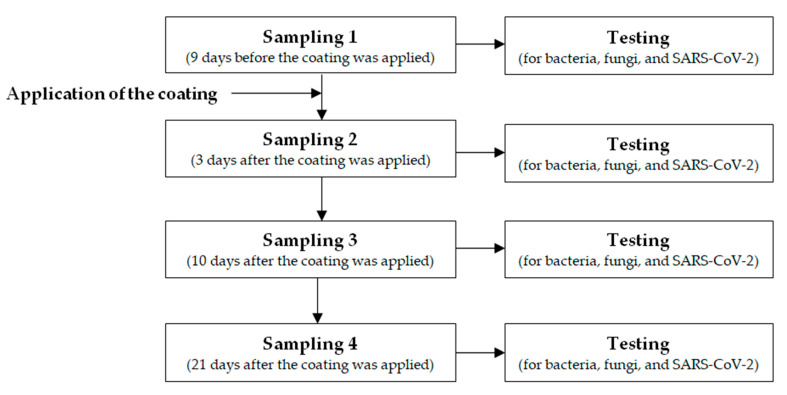
Flow of environmental investigation.

**Table 1 ijerph-20-05434-t001:** Samples tested positive for bacteria in Hospital A by sampling site and sampling phase.

Sampling Site	Sampling 1 *	Sampling 2 *	Sampling 3 *	Sampling 4 *
Emergency Department	12/15	0/15	0/15	0/15
COVID-19 patients’ rooms	6/8	0/8	0/8	0/8
Intensive Care Unit	12/20	2/20	2/20	1/20
Radiology and CT Department	10/12	2/12	0/12	0/12
Stairs and elevators	13/14	0/14	1/14	0/14

* sampling 1:9 days before the coating was applied; sampling 2:3 days after the coating was applied; sampling 3:10 days after the coating was applied; sampling 4:21 days after the coating was applied. COVID-19: coronavirus disease 2019; CT: computed tomography.

**Table 2 ijerph-20-05434-t002:** Samples tested positive for fungi in Hospital A by sampling site and sampling phase.

Sampling Site	Sampling 1 *	Sampling 2 *	Sampling 3 *	Sampling 4 *
Emergency Department	3/15	0/15	0/15	0/15
COVID-19 patients’ rooms	0/8	0/8	0/8	0/8
Intensive Care Unit	3/20	0/20	0/20	0/20
Radiology and CT Department	1/12	0/12	0/12	0/12
Stairs and elevators	2/14	0/14	0/14	0/14

* sampling 1:9 days before the coating was applied; sampling 2:3 days after the coating was applied; sampling 3:10 days after the coating was applied; sampling 4:21 days after the coating was applied. COVID-19: coronavirus disease 2019; CT: computed tomography.

**Table 3 ijerph-20-05434-t003:** Samples tested positive for SARS-CoV-2 by hospital, sampling site, and sampling phase.

Hospital A	Sampling 1 *	Sampling 2 *	Sampling 3 *	Sampling 4 *
Emergency Department	0/15	0/15	0/15	0/15
COVID-19 patients’ rooms	1/8	0/8	0/8	0/8
Intensive Care Unit	0/20	0/20	0/20	0/20
Radiology and CT Department	1/12	0/12	0/12	0/12
Stairs and elevators	0/14	0/14	1/14	0/14
**Hospital B**	**Sampling 1 ***	**Sampling 2 ***	**Sampling 3 ***	**Sampling 4 ***
Emergency Department	3/19	0/19	0/19	0/19
COVID-19 patients’ rooms	2/6	0/6	0/6	0/6
Intensive Care Unit	0/20	0/20	0/20	0/20
Radiology and CT Department	1/11	0/11	0/11	0/11
Stairs and elevators	2/14	0/14	0/14	0/14

* sampling 1:9 days before the coating was applied; sampling 2:3 days after the coating was applied; sampling 3:10 days after the coating was applied; sampling 4:21 days after the coating was applied. SARS-CoV-2: severe acute respiratory syndrome coronavirus 2; COVID-19: coronavirus disease 2019; CT: computed tomography.

## Data Availability

Data are available upon permission request.

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
