# Peer review of "Effectiveness of a Self-Decontaminating Coating Containing Usnic Acid in Reducing Environmental Microbial Load in Tertiary-Care Hospitals"

_ijerph, 2023, doi:10.3390/ijerph20085434_

Round 1

Reviewer 1 Report

Thank you for inviting me to review this work entitled "Effectiveness of a self-decontaminating coating containing us- 2 nic acid-containing in reducing environmental microbial load 3 in tertiary-care hospitals," which aimed to estimate the effectiveness of a usnic acid-containing self-decontaminating coating in reducing surface contamination in tertiary-care hospitals. 

Some comments have been raised:

1. Abstract: please one sentence about the clinical implication of the study findings

2. Introduction: very well written

3. Methods: Written in detail (which is appreciated); however, I still had difficulty to follow the track of work> I would recommend adding a scheme or info-graphical figure to illustrate the phases. 

4. Results: this section needs elaboration, hence most of the data were presented in tables. 

5. the citation of table 4 before table 3? re-number, please

6. Tabel number 4 is a little bit misleading; the reader will be able to understand the importance of the findings. I would recommend keeping these details as an appendix and re-present the most important outcomes in figures of tables. 

Author Response

Comments and Suggestions for Authors

Thank you for inviting me to review this work entitled "Effectiveness of a self-decontaminating coating containing us- 2 nic acid-containing in reducing environmental microbial load 3 in tertiary-care hospitals," which aimed to estimate the effectiveness of a usnic acid-containing self-decontaminating coating in reducing surface contamination in tertiary-care hospitals. 

Some comments have been raised:

Comment 1

Abstract: please one sentence about the clinical implication of the study findings

Answer to Comment 1

We added a sentence to indicate the clinical implication of our findings (last sentence of Abstract). Thank you! Please also note that our manuscript was reviewed and edited by a native-English individual.

Comment 2

Introduction: very well written

Answer to Comment 2

Thank you!!

Comment 3

Methods: Written in detail (which is appreciated); however, I still had difficulty to follow the track of work> I would recommend adding a scheme or info-graphical figure to illustrate the phases. 

Answer to Comment 3

As recommended, we added Figure 1 in order to facilitate the readers. Thank you!

Comment 4

Results: this section needs elaboration, hence most of the data were presented in tables. 

Answer to Comment 4

As requested, more information was added in the Results section to present in details our findings. 

Comment 5

the citation of table 4 before table 3? re-number, please

Answer to Comment 5

We thank the reviewer for the detailed comments on our manuscript. Please note that in the first sentence of Results section (page 4) we mention gthe first 3 Tables (Table 1, Table 2, Table 3). Therefore, Table 4, is not cited before Table 3.

Comment 6

Tabel number 4 is a little bit misleading; the reader will be able to understand the importance of the findings. I would recommend keeping these details as an appendix and re-present the most important outcomes in figures of tables. 

Answer to Comment 6

We totally agree with this comment. For this reason, we resubmitted Table 4 as Supplementary File 1. Details about the findings of the (new) Supplementary File 1 are already shown in the manuscript.   

Reviewer 2 Report

Following are the suggestions to improve the paper.

1) Introduction needs extensive literature review that what are the needs of the study and what is the gap. I have not found the gap of study. Further what type of work already published and how you contribute in this topic.

2) Method is good but statistic was not according to the level.

3) Limitation of the study is missing.

4) Inclusion and exclusion criteria is also missing.

5) Conclusion is not actually representing the results.

Author Response

Comments and Suggestions for Authors

Following are the suggestions to improve the paper.

Comment 1

Introduction needs extensive literature review that what are the needs of the study and what is the gap. I have not found the gap of study. Further what type of work already published and how you contribute in this topic.

Answer to Comment 1

We thank the reviewer for the valuable comments to improve our manuscript. We clarified in the Abstract (2nf and 3rd line) the aim of this study. Furthermore, we clarified the aim of this study at the last paragraph of Introduction (page 2). In our opinion, we presented in details the gaps on this issue, as well as the relevant literature (30 articles only in Introduction). Given that there are no published data on the use of an usnic-acid containing coating in healthcare facilities; this was clarified in Introduction (page 2, last paragraph). Please also note that a native English-speaker reviewed and edited our revised manuscript.

Comment 2

Method is good but statistic was not according to the level.

Answer to Comment 2

Thank you for your comment but we are a bit unsure what you meant with this. We believe that the statistical method is described well and it is according to the study aim to estimate the effectiveness (estimate the reduction) of the coating in reducing microbial surface contamination. There was the following sentence in the methods before, which we have now removed to facilitate the reader. Apologies for that. “The statistical significant reduction of the microbial load indicated high effectivity, while good antimicrobial activity was defined as the activity that fulfills most of the properties described by the Centers for Disease Control and Prevention [33]”.

It would be great if you can clarify if your comment was regarding this sentence or otherwise elaborate your comment in order to assist us to address it in the next revised draft.   

Comment 3

Limitation of the study is missing.

Answer to Comment 3

There is a limitation section in Discussion (the paragraph before Conclusions). However, we added more on this issue in order to expand this section (e.g., Another potential limitation is that the subsequent collection of two samples from each surface could limit the microbial load on the tested surface.). Thank you! 

Comment 4

Inclusion and exclusion criteria is also missing.

Answer to Comment 4

To our knowledge, there are no inclusion and exclusion criteria in our study. Since there is a possibility that we did not understand this suggestion, please provide more details for this comment so we can address it. Thank you!  

Comment 5

Conclusion is not actually representing the results.

Answer to Comment 5

We thank the reviewer for this particular comment. Indeed, we revised the whole section of Conclusions in order to comply with the reviewer’s comment (page 10).

Reviewer 3 Report

The authors demonstrate in their manuscript the antimicrobial efficacy of a usnic acid-containing coating in a hospital setting. The authors are suggesting that the coating is effective against multidrug-resistant (MDR) pathogens; see line 50-54 and 267-271. Such has not been studied. Therefore they should indicate that the coating is effective against contamination with pathogens and perhaps against MDR pathogens. 

Points to consider:

Line 38: delete word "highly". 

Line 39: add in last part of sentence "contamination on surfaces in hospitals".

Line 117, 133, 191, 199, 205, 210, and 243: 102 should be written as 100

Line 117: ISO 18593 was followed. Since you have to buy the standard, please summarize how samples were taken.

Line 126 and 127: substitute "elusion" by elution.

Line 130: substitute "Clostridium" by Clostridioides

Line 131:  "whole plate agar"; what does it mean?; Please indicate what agar media are used, what incubation conditions are used. 

Line 133: "Pathogenic and ubiquitous microorganisms" .....evaluated the bacterial growth"... This is a bit strange. Apparently, you considered the total bacterial presence. 

Table 3: locate table on one page.

Table 4; line 180: "Isolated bacteria", these are identified; how?MicroScan autoSCAN-4 System (line 124)? describe briefly in materials and methods. Now it states "detected with" and actually it should be identified. Correct?

With regard to Table 4, is there a trend you observe that explains the location of a particular species? If not and if not discussed, then the table is obsolete. Just mention what species were encountered. 

Line 218-219: "These findings indicate high effectivity of the coating"; remove "high" and add instead "antimicrobial"

Line 245: about the microorganisms that were encountered; how about C. difficile?

Line 257: "detection does not imply infectiousness"; true, but what are detection levels. Did you use a quantitative RT-PCR? Perhaps discuss here also how long infectiousness of SCoV-2 remains when it is deposited on surfaces.

Line 262-263: Indicate that a follow-up study would be necessary to demonstrate the antimicrobial life-time of the coating.

Line 268: MDR contamination has not been specifically studied. So it's relevant to explain how it contributes. MDR pathogens may be more resistant to usnic acid. Such would require a different approach. 

Line 270: delete the word "very".

Line 271-273: rephrase the sentence! The data support the benefit of usnic acid coating in reducing the microbial load on healthcare surfaces.

Author Response

Comments and Suggestions for Authors

The authors demonstrate in their manuscript the antimicrobial efficacy of a usnic acid-containing coating in a hospital setting. The authors are suggesting that the coating is effective against multidrug-resistant (MDR) pathogens; see line 50-54 and 267-271. Such has not been studied. Therefore they should indicate that the coating is effective against contamination with pathogens and perhaps against MDR pathogens. 

Answer to Comment

We total agree with this comment. For this reason, we excluded the sentence referring to the call the WHO (Conclusions, line 267-271). Please read our answer to Point 16. Please note that regarding English language, our manuscript has been reviewed and edited by a native English-speaker. Thank you!   

Points to consider:

Point 1

Line 38: delete word "highly". 

Answer to Point 1

We deleted word “highly”, as recommended. Thank you!

Point 2

Line 39: add in last part of sentence "contamination on surfaces in hospitals".

Answer to Point 2

We added, as recommended. Thank you! 

Point 3

Line 117, 133, 191, 199, 205, 210, and 243: 102 should be written as 100

Answer to Point 3

We corrected accordingly, thank you!!

Point 4

Line 117: ISO 18593 was followed. Since you have to buy the standard, please summarize how samples were taken.

Answer to Point 4

Please note that the detailed description of samples collection is already presented in the Environmental Investigation section (page 3).

Point 5

Line 126 and 127: substitute "elusion" by elution.

Answer to Point 5

We substituted as recommended, thank you.

Point 6

Line 130: substitute "Clostridium" by Clostridioides

Answer to Point 6

We corrected, thank you!

Point 7

Line 131:  "whole plate agar"; what does it mean?; Please indicate what agar media are used, what incubation conditions are used. 

Answer to Point 7

In response to reviewer comment, the “Laboratory section” (Methods, page 4) was clarified as follows: Elution medium swabs were cultured on conventional solid media for a wide range of Gram-positive and Gram-negative bacteria and fungi: blood agar, chocolate agar, MacConkey agar, Sabouraud – dextrose agar, mannitol – salt agar (Chapman), and Clostridioides difficile agar. For each swab, 200 μL of elution medium was streaked on a whole agar plate. Colonies were counted after incubation for 24-48 h. Incubation was performed at 37o C under aerobic conditions for all culture media, except for C. difficile that was incubated at 37o C under anaerobic conditions.

Point 8

Line 133: "Pathogenic and ubiquitous microorganisms" .....evaluated the bacterial growth"... This is a bit strange. Apparently, you considered the total bacterial presence. 

Answer to Point 8

We agree with this comment. We corrected accordingly.  

Point 9

Table 3: locate table on one page.

Answer to Point 9

Done! Thank you!

Point 10

Table 4; line 180: "Isolated bacteria", these are identified; how?MicroScan autoSCAN-4 System (line 124)? describe briefly in materials and methods. Now it states "detected with" and actually it should be identified. Correct?

Answer to Point 10

We thank the reviewer for pointing this out. We corrected the title of (old) Table 4 (now Supplementary File 1) accordingly. We also corrected the word “detected” to “identified” in Methods (page 4, Laboratory testing section).     

Point 11

With regard to Table 4, is there a trend you observe that explains the location of a particular species? If not and if not discussed, then the table is obsolete. Just mention what species were encountered. 

Answer to Point 11

Following the recommendations of Reviewer 1, we resubmitted Table 4 as a Supplementary File. Thank you for your comment!  

Point 12

Line 218-219: "These findings indicate high effectivity of the coating"; remove "high" and add instead "antimicrobial"

Answer to Point 12

We corrected the sentence, as recommended. Thank you!

Point 13

Line 245: about the microorganisms that were encountered; how about C. difficile?

Answer to Point 13

There was no growth of C. difficile in any of the samples in our study.

Point 14

Line 257: "detection does not imply infectiousness"; true, but what are detection levels. Did you use a quantitative RT-PCR? Perhaps discuss here also how long infectiousness of SCoV-2 remains when it is deposited on surfaces.

Answer to Point 14

The Real-time RT-PCR used for the detection of SARS-CoV-2 virus is qualitative. This information was added in Laboratory Testing section (page 4). Regarding the duration of infectiousness when the virus is deposited on surfaces, we added the following text in Discussion (page 10) and we also used 2 more references (new  references 42 and 43). 

“Previous studies have shown that the virus is able to survive on various porous and non-porous surfaces like glass, steel or certain types of plastic [42], for up to 28 days at 20°C [43]”.

References:

[42] Liu, Y., Li, T., Deng, Y., Liu, S., Zhang, D., Li, H., … Li, J. (2021). Stability of SARS-CoV-2 on environmental surfaces and in human excreta. Journal of Hospital Infection, 107, 105–107.

[43] Geng Y. & Wang Y.(2022). Stability and transmissibility of SARS-CoV-2 in the environment. Journal of Medical VirologyVolume 95, Issue 1.

Point 15

Line 262-263: Indicate that a follow-up study would be necessary to demonstrate the antimicrobial life-time of the coating.

Answer to Point 15

We agree with this particular comment. We added the sentence, as recommended. 

Point 16

Line 268: MDR contamination has not been specifically studied. So it's relevant to explain how it contributes. MDR pathogens may be more resistant to usnic acid. Such would require a different approach. 

Answer to Point 16                                          

We agree with this particular comment. For this reason, we deleted the sentence, in order to avoid confusing the reader. Thank you!  

Point 17

Line 270: delete the word "very".

Answer to Point 17

We deleted the word “very”, thank you.

Point 18

Line 271-273: rephrase the sentence! The data support the benefit of usnic acid coating in reducing the microbial load on healthcare surfaces.

Answer to Point 18

We revised the sentence, as requested. Thank you!

Round 2

Reviewer 1 Report

Thank you for addressing my comments. I believe it is publishable now

Reviewer 2 Report

After revision, Paper may be accepted. Introduction , method, result and discussion section is good and accordingly.